# Undernutrition combined with dietary mineral oil hastens depuration of stored dioxin and polychlorinated biphenyls in ewes. 2. Tissue distribution, mass balance and body burden

Sylvain Lerch[1,2]*, Lucille Rey-Cadilhac[1,3], Ronan Cariou[4], Yannick Faulconnier[3], Catherine Jondreville[1], Denis Roux[5], Gaud Dervilly-Pinel[4], Bruno Le Bizec[4], Stefan Jurjanz[1], Anne Ferlay[3]

1 UR AFPA, Université de Lorraine, INRAE, Nancy, France, 2 Ruminant Research Unit, Agroscope, Posieux, Switzerland, 3 UMR Herbivores, Université Clermont Auvergne, INRAE, VetAgro Sup, Saint-Genès-Champanelle, France, 4 LABERCA, Oniris, INRAE, Nantes, France, 5 UE Herbipôle, INRAE, Saint-Genès-Champanelle, France

* sylvain.lerch@agroscope.admin.ch

**Data Availability Statement:** Detailed individual data are available in the data.inra.fr repository

## Abstract

Food safety crises involving persistent organic pollutants (POPs) lead to systematic slaughter of livestock to prevent contaminants from entering the food chain. Therefore, there is a need to develop strategies to depurate livestock moderately contaminated with POPs to reduce economic and social damage. This study aimed to test undernutrition (37% of energy requirements) combined with mineral oil (10% in total dry matter intake) in nine non-lactating ewes contaminated with 2,3,7,8-tetrachlorodibenzo-*p*-dioxin (TCDD) and polychlorinated biphenyls (PCBs) 126 and 153 as a strategy to enhance the depuration of POPs through faecal excretion. To better understand the underlying mechanisms of the depuration process, lipophilic POPs and lipid fluxes were co-monitored in various body and excretion compartments. Body compartments (adipose tissues, muscle, liver and blood) and the total empty body were analyzed for lipids and POPs concentrations and burdens at slaughter, as well as excretion compartments (faeces and wool) collected during the depuration period. Decreases in empty body total and lipid weights were 6-fold higher in underfed and supplemented ewes compared to control ewes. In addition, over the depuration period undernutrition and supplementation treatment increased faecal TCDD, PCBs 126 and 153 excretions by 1.4- to 2.1-fold but tended to decrease wool PCB 153 excretion by 1.4-fold. This induced 2- to 3-fold higher decreases in the empty body POPs burdens for underfed and supplemented ewes. Nonetheless, when expressed relative to the calculated initial empty body burdens, burdens at slaughter decreased only slightly from 97%, 103% and 98% for control ewes to 92%, 97% and 94% for underfed and supplemented ones, for TCDD, PCBs 126 and 153, respectively. Fine descriptions at once of POPs kinetic (companion paper 1) and mass balance (companion paper 2), and of body lipid dynamics were very useful in improving our understanding of the fate of POPs in the ruminants.

(open access at https://doi.org/10.15454/Z6UML7).

**Funding:** This work was supported by the "Conseil Régional de Lorraine" (Nancy, France) under Grant "Université/EPST-Région 2014" and by the "Centre National Interprofessionnel de l'Economie Laitière" (CNIEL, Paris, France). The funders had no role in study design, data collection and analysis, decision to publish, or preparation of the manuscript.

**Competing interests:** The authors have declared that no competing interests exist.

## Introduction

Persistent Organic Pollutants (POPs) are chemical molecules that are toxic for wildlife and humans. They have specific physico-chemical properties: they are persistent in the environment and bioaccumulate in animal fat-rich tissues, organs and excreta due to their lipophilicity [1]. Accidental contamination of livestock by POPs has occurred in past decades, resulting in POPs concentrations in animal products that exceeded the maximum regulatory levels [2]. When such incidents happen, the contaminated livestock and their food products are often disposed rather than saved, as the depuration process for POPs is extremely slow. Therefore, there is a real need to develop rearing strategies to depurate contaminated animals to reduce the deleterious social and economic damage caused by food contamination crises involving POPs.

Due to their lipophilic nature, the fate of POPs in animal organism is thought to be linked to the dynamics of body lipids. Based on this assumption, the aim of the present study is to assess the effectiveness of undernutrition (inducing the mobilization of body fat reserves) combined with mineral oil (MO) supplementation (allowing increased faecal lipid excretion) as a strategy to decontaminate ewes from stored 2,3,7,8-tetrachlorodibenzo-$p$-dioxin (TCDD), and polychlorobiphenyls (PCBs) 126 and 153. These three molecules were chosen as representatives of highly persistent POPs characterized by poor metabolization and a low depuration rate in the milk and meat of ruminants among the dioxins (TCDD), dioxin-like PCBs (PCB 126) and non dioxin-like PCBs (PCB 153) [3–5].

The first companion paper focuses on the kinetics of POPs concentrations in the main storage tissue (adipose tissue; AT), the central distribution compartment (blood), and the excretion pool (faeces), and shows that undernutrition combined with MO supplementation efficiently increases the faecal concentration of POPs and decreases the pericaudal subcutaneous AT burden [6]. The rationale of this second companion paper is to further explore the POPs tissue distribution process and to quantitatively and precisely measure both faecal and wool excretion patterns and their consequences on the dynamics of body burdens. Indeed, quantitatively at the whole animal scale, a 3-fold decrease in the total body burden of PCBs was observed after 21 days of undernutrition combined with MO supplementation in growing chickens [7]. In comparison, severe undernutrition promoted a redistribution process of dichlorodiphenyltrichloroethane (DDT) from AT to oxidative muscles in pigeons [8] and cockerels [9]. Conversely to birds, the effect of undernutrition combined with dietary non-absorbable lipid supplementation on quantitative faecal and dander excretions, body burden, and tissue distribution remains unknown in ruminants, while differential responses should be expected due to differences in the body size, digestion, physiology, lipid nutrition and metabolism of ruminants when compared to birds [10].

## Materials and methods

### Ethics statement

All experimental procedures were approved by the French Ministry of Research and Higher Education (agreement n° 2357–20151008171318) after an ethical evaluation by the committee C2EA-02 (Clermont-Ferrand, France). The number of animals per group was determined according to an *a priori* experimental power test performed before starting the experiment, as described in the first companion paper [6] and according to the expected intra-group variability based on previous experiments and the literature [11, 12]. *A posteriori*, significant differences ($P \leq 0.05$) between treatments for tissue POPs concentrations were observed, which confirms that the number of animals used was sufficient and properly estimated *a priori*.

## Animals and diets

The experimental design was described in detail in the first companion paper [6]. In brief, nine non-lactating Romane ewes were used during three successive periods: i) a 27-day POPs exposure period during which ewes were feedly exposed to TCDD, PCBs 126 and 153 through a spiked concentrate; ii) an 8-day buffering period with a "clean" diet; and iii) a 58-day depuration period when ewes were allocated into two groups and received one of two depuration treatments. Four ewes received a control well-fed and non-supplemented treatment (CTL) covering 96% of maintenance energy requirements (MER), and five ewes received an underfed (37% of MER) and MO supplemented [10% of total dry matter (DM) intake] treatment (UFMO). Diets were composed of barley straw, hay and dehydrated beet pulp/corn grain-based concentrate.

## Sampling and measurements

**Feed.** The amounts of each of the distributed feedstuffs were weighed individually and daily as well as refusals, if any. Subsamples of feedstuffs were collected weekly, pooled for the entire experiment, and ground through a 1-mm sieve before subsequent chemical analyses.

**Faeces.** Every 3–4 days over the depuration period, 50 g to 150 g of fresh faeces were individually collected straight from the rectum at approximately 0800 h (16 samples per ewe). Faecal samples were stored at -20˚C and lyophilized before one pool per ewe of dry faeces was composited based on proportional amounts depending on the DM intake measured on the five days preceding each sampling. Nine individual faecal pools representative of faecal excretion over the depuration period were thus obtained and ground through a 1-mm sieve before chemical analyses. Faecal outputs were further estimated using acid-insoluble ashes as an indigestible marker [13] analyzed in pools of feedstuffs and individual faeces.

**Wool.** Ewes were shorn to a wool length of 0.5 cm to 1 cm on Day -1 and Day +56 of the depuration period. An area of approximatively 200 cm$^2$ located on the upper left side of the body was shaved further until the skin was completely free of wool. On Day +56, all the fleece of each ewe was individually weighed to estimate wool growth during the depuration period. The wool from the area that was totally shaved was stored at -20˚C, lyophilized, and finely ground in liquid nitrogen using a ball mill, before chemical analyses.

**Body tissues.** At the end of the depuration period (Day +57) at 0800 h before feed distribution, blood was collected by venipuncture from the jugular vein in tubes containing a clot activator (SiO$_2$) and maintained at 4˚C for 20 h. Serum was then separated by centrifugation (1 400 *g*, 25 min, 20˚C) and kept at -80˚C before chemical analyses. Immediately after blood sampling, ewes were weighed and body condition score was estimated on a 0 to 5 scale [14] before slaughter between 0900 and 1000 h by stunning, followed by exsanguination. Subsamples of approximately 50 mg of mesenteric, perirenal, and pericaudal subcutaneous AT were collected for adipocyte measurements. The rest of the three AT together with liver and *Rectus abdominis* muscle were exhaustively collected, weighed, stored at -20˚C, lyophilized, and finely ground in liquid nitrogen using a blade mill before chemical analyses.

Subsequently, gut contents were exhaustively removed from the digestive tract before the empty body (whole body including blood but minus gut contents and wool) was cut into five or six pieces. All the collected parts of the empty body were stored at -20˚C in hermetic plastic bags. The contents of each bag were weighed before and after removal from cold storage, and any losses observed since the initial weighing were assumed to be water. For each ewe, the frozen contents of the bags (including blood and the emptied digestive tract but excluding exhaustive samples of AT, liver and muscle) were minced, mixed, and homogenized at the "UE Porcs Rennes" (INRA Saint-Gilles, France). This was done using an industrial flaker

(Rotary Meat Flaker, model RF15; Hobart, Cesson-Sévigné, France) to render meat blocks into a size suitable for grinding and an industrial mixer–grinder (model 4346; Hobart) to grind and homogenize. A homogenized 1-kg aliquot was obtained, stored at -20˚C, lyophilized, and finely ground in liquid nitrogen using a blade mill before chemical analyses.

## Chemical analyses

Feedstuffs and faeces were analyzed for DM, total ashes, total lipids [15] and acid-insoluble ashes [13]. For wool and body compartments, DM was determined by lyophilisation, ashes by calcination (550˚C furnace for 6 h) and total lipids gravimetrically after chloroform/methanol extraction [16], except for blood serum where lipids were quantified by spectrophotometry using a commercial kit (HB018; Cypress Diagnostics, Langdorp, Belgium). Adipocyte cell measurements were performed for each AT depot after immediate collection at slaughter in physiological saline at 39˚C for less than 1 h, followed by fixation in osmium oxide tetroxide for at least one week [17]. The arithmetic means of diameter and volume of 350 to 450 fixed adipocytes with diameters $\geq$ 25 μm were then determined under microscope.

Concentrations of TCDD, PCBs 126 and 153 in feedstuffs, body compartments, faeces and wool were determined according to ISO/IEC 17025:2005 fully accredited methods (except for faeces and wool) that were slightly adapted from previous methods [18, 19]. In terms of the wool extraction part, the method was adapted from Nakao et al. [20]. The methods are detailed in the S1 File.

## Calculations and statistical analyses

Empty body weight was computed by subtracting the weight of gut contents from body weight (BW) at slaughter. The empty body weights of DM, lipids, ashes, and the TCDD and PCBs empty body burdens were computed by summing the amounts of each chemical component in every body compartment collected and analyzed at slaughter (i.e., mesenteric, perirenal and pericaudal subcutaneous AT, liver, *Rectus abdominis* muscle and the rest of the empty body). As proposed by Atti and Bocquier [21], allometric equations were adjusted between the natural logarithms of the weight of empty body lipids and the weight of the three AT and *Rectus abdominis* muscle lipids:

Log (individual AT or muscle lipids weight, g) = $a \times$ log (empty body lipids weight, kg)$-b$
Where the slope of the relationship $a$ is called the allometric coefficient.

Assimilation efficiencies of POPs before the depuration treatment were also calculated as the estimated body burdens of the POPs at the beginning of the depuration period (Day 0) divided by the cumulative POPs feed inputs during the exposure and buffering periods (Day -35 to Day -1).

Data were analyzed by ANOVA using the MIXED procedure of SAS (SAS 9.3., 2003). Total and empty body weights and composition, body condition score, adipocyte measurements, amount of POPs intake or excreted through faeces or wool, and POPs burdens of empty body and individual tissues were analyzed separately for Day 0 and Day +57 using a statistical model that included depuration treatment (CTL and UFMO) as a fixed effect and ewe as a random effect. Concentrations of POPs for all body tissues measured at slaughter (Day +57) and faeces and wool concentrations during the depuration period (Day 0 to Day +57) were analyzed together using a statistical model that included depuration treatment, compartment (body or outputs compartments) and treatment by compartment interaction as fixed effects, ewe as a random effect, and compartment as a repeated statement. An autoregressive first-order covariance structure was used. When the treatment by compartment interaction was significant, treatment differences for each compartment were determined based on the results of the slice

option of SAS. Treatment differences were determined based on *t*-tests and declared significant at $P \leq 0.05$. Trends towards significance were considered at $0.05 < P \leq 0.10$. Values reported are least square means and standard error of the mean. Codes for statistical analyses are provided in the S2 File.

## Results

### Body weight and composition

Body weight, empty body total and lipid weights, body condition score and adipocyte diameters are presented in Table 1. Details about the empty body chemical composition are presented in S1 and S2 Tables.

At the end of the depuration period, the body condition score was lower ($P = 0.01$) and total empty body weight tended to be lower ($P = 0.10$) for UFMO ewes compared to CTL ewes. Moreover, over the 57 days of the depuration period, decreases in BW, empty body total and lipid weights, and body condition score were on average 6-fold higher ($P < 0.01$) for UFMO compared to CTL treatment. The adipocyte diameter remained unaffected by treatment at the end of the depuration period whatever the AT, even if decrease over the depuration period in pericaudal subcutaneous adipocyte diameter was significant for UFMO ewes (-12 μm, $P < 0.05$; Table 1). Compared to the CTL treatment, the UFMO treatment had no effect on the weights of any empty body compartment studied, except for subcutaneous AT protein, *Rectus abdominis* muscle ashes and serum lipids ($P < 0.05$; S2 Table). At slaughter, calculated allometric coefficients for body tissue lipid weight compared to total empty body lipid weight were higher ($P < 0.05$) for perirenal (2.6) and pericaudal subcutaneous (2.4) AT than for mesenteric AT (0.9) and *Rectus abdominis* muscle (0.4).

### Body tissues, faeces and wool POPs concentrations

Body tissues, faeces and wool POPs concentrations expressed per g lipids are reported in Fig 1 and S3 Table.

**Table 1. Anatomical measurements, body fatness and adipocyte diameters of depurated ewes[1].**

| Item | Day of depuration period[1] | | | | | | | | Changes during depuration period Day +57 minus Day 0 | | | |
|---|---|---|---|---|---|---|---|---|---|---|---|---|
| | Day 0 | | | | Day +57 | | | | | | | |
| | Treatment | | SEM | *P*-value | Treatment | | SEM | *P*-value | Treatment | | SEM | *P*-value |
| | CTL | UFMO | | | CTL | UFMO | | | CTL | UFMO | | |
| BW (kg) | 62.0 | 64.2 | 2.8 | 0.60 | 59.4 | 50.6 | 5.6 | 0.31 | -2.6 | -13.6* | 1.4 | <0.01 |
| Empty body weight (kg)[2] | 50.3 | 52.2 | 2.6 | 0.63 | 47.8 | 40.9 | 2.7 | 0.10 | -2.5 | -11.3* | 1.7 | <0.01 |
| Empty body lipids (kg)[2] | 12.6 | 14.7 | 1.2 | 0.26 | 11.6 | 9.5 | 1.6 | 0.39 | -1.0 | -5.2* | 0.5 | <0.01 |
| Body condition score (0 to 5) | 3.6 | 3.6 | 0.2 | 0.92 | 3.5 | 2.6 | 0.2 | 0.01 | -0.1 | -1.0* | 0.1 | <0.001 |
| Adipocyte diameter (μm) | | | | | | | | | | | | |
| Mesenteric | *Not determined* | | | | 93 | 88 | 4 | 0.38 | - | | | |
| Perirenal | *Not determined* | | | | 98 | 85 | 10 | 0.37 | - | | | |
| Pericaudal sc.[3] | 78 | 83 | 5 | 0.52 | 75 | 71 | 4 | 0.58 | -4 | -12* | 6 | 0.34 |

[1]Four ewes received a control well-fed and non-supplemented treatment (CTL), while five ewes received an underfed and mineral oil supplemented treatment (UFMO).

[2]Empty body: total body minus gut contents and wool. *Post-mortem* measurements for Day+57 or estimated at Day 0: for empty body weight based on measured BW and DM intake, considering that gut content weight is equal to five times DM intake [22], and for empty body lipid weight based on decreases in BW between Day 0 and Day +57 and considering -0.38 kg body lipids for -1 kg BW [23].

[3]Pericaudal sc.: Pericaudal subcutaneous.

*Indicates that least square mean for change over the depuration period is different from 0 at *P*<0.05.

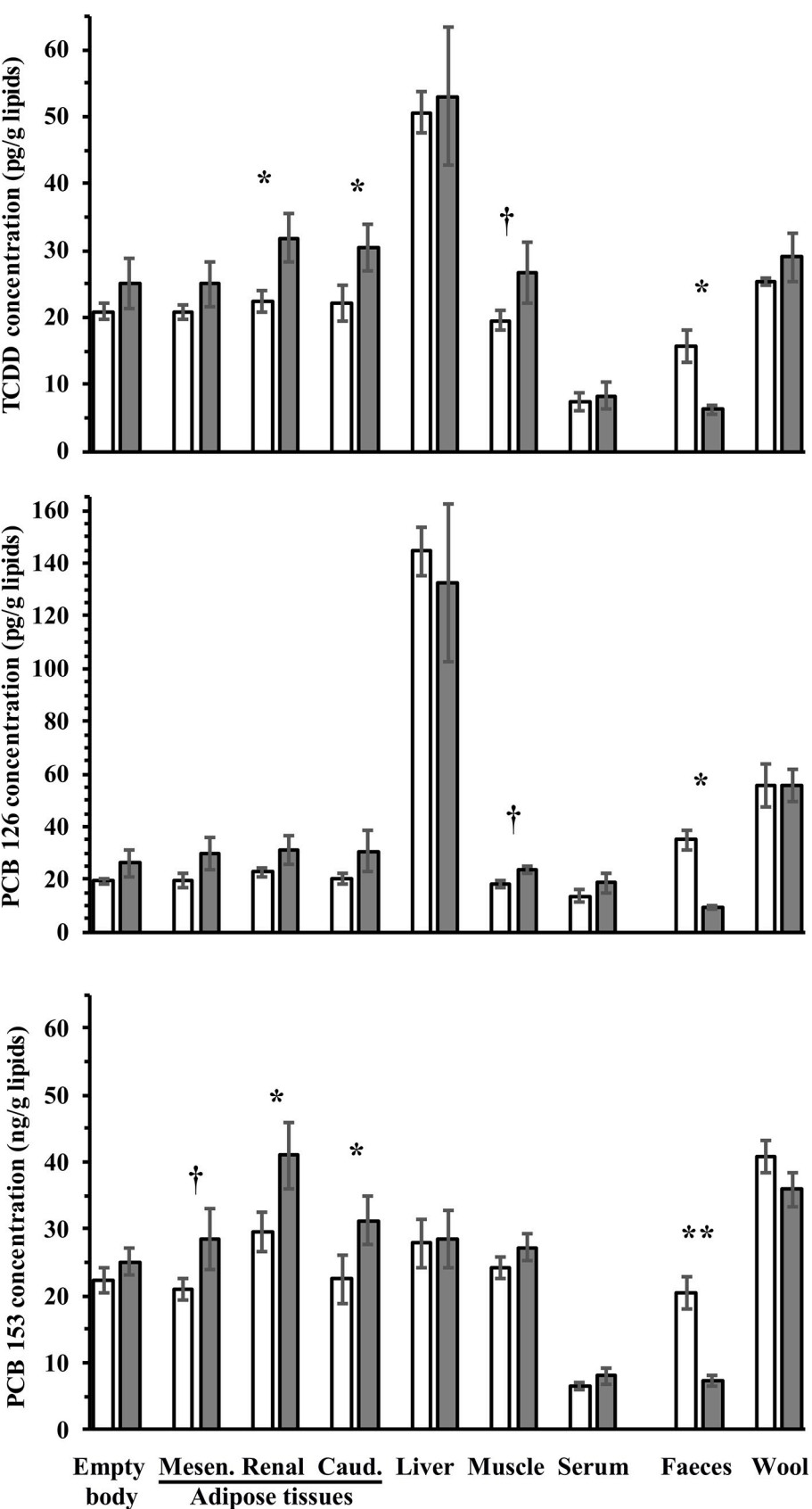

**Fig 1. Pollutant concentrations in body tissues of ewes at slaughter and in excretion compartments over the depuration period in ewes receiving a control well-fed and non-supplemented treatment (CTL, open bars; _n_ = 4), or an underfed and mineral oil supplemented treatment (UFMO, grey bars; _n_ = 5).** Bars represent least-squares means, error bars represent SEM, * and ** represent a difference ($P \leq 0.05$ and 0.01, respectively) and † represents a tendency for difference ($P \leq 0.10$) between treatments. Mesen.: mesenteric; Caud.: pericaudal subcutaneous.

Depuration treatment affected the concentrations of POPs in various compartments depending on the treatment by compartment interaction. Compared to CTL, UFMO treatment increased ($P \leq 0.05$) perirenal and pericaudal subcutaneous AT TCDD and PCB 153 concentrations and tended to increase ($P \leq 0.10$) muscle TCDD and PCB 126, and mesenteric AT PCB 153 concentrations. Conversely, UFMO treatment decreased ($P \leq 0.05$) faeces concentrations of TCDD, PCBs 126 and 153. Concentrations also varied widely between body tissues or output compartments whatever the treatment. Thus, for both CTL and UFMO treatments, liver TCDD and PCB 126 concentrations were the highest ($P \leq 0.05$) and serum TCDD and PCB 153 concentrations were the lowest ($P < 0.01$) across all body tissues, whereas wool concentrations of PCBs 126 and 153 were higher ($P < 0.05$) compared to empty body, AT and muscle concentrations. Moreover, in UFMO ewes, perirenal AT TCDD and PCB 153 concentrations were higher ($P < 0.05$) than empty body, mesenteric AT and muscle concentrations (Fig 1 and S3 Table).

## Body tissue burdens and faecal and wool outputs of POPs

Empty body burdens, cumulated feed inputs, and faecal and wool outputs for TCDD and PCBs are reported in Table 2, and detailed body tissue burdens are presented in Table 3.

In addition, the TCDD and PCBs input/output balances are graphically reported in Fig 2. Assimilation efficiencies at the beginning of the depuration period were equal to 50%, 48%

**Table 2. Pollutants inputs / outputs / body burdens mass balances[1].**

| | Oral inputs by period | | | Outputs—Depuration period (Days 0 to +57) | | Empty body[2] burdens—Depuration period | |
|---|---|---|---|---|---|---|---|
| | Exposure Days -35 to -9 | Buffering Days -8 to -1 | Depuration Days 0 to +57 | Faeces | Wool | Day 0 | Day+57 |
| TCDD (ng) | | | | | | | |
| CTL | 474 | _Not detected_ | _Not Detected_ | 8.0 | 0.39 | 247 | 239 |
| UFMO | 486 | | | 16.8 | 0.35 | 233 | 216 |
| SEM | 23 | | | 1.9 | 0.06 | 19 | 19 |
| _P_-value | 0.72 | | | 0.02 | 0.67 | 0.62 | 0.43 |
| PCB 126 (ng) | | | | | | | |
| CTL | 481 | 3.5 | 25.2 | 18.0 | 0.80 | 220 | 227 |
| UFMO | 494 | 3.9 | 20.7 | 25.7 | 0.68 | 241 | 236 |
| SEM | 23 | 0.3 | 1.5 | 2.3 | 0.09 | 42 | 41 |
| _P_-value | 0.70 | 0.30 | 0.07 | 0.05 | 0.41 | 0.73 | 0.88 |
| PCB 153 (µg) | | | | | | | |
| CTL | 475 | 0.9 | 6.7 | 10.6 | 0.61 | 258 | 253 |
| UFMO | 487 | 1.0 | 4.7 | 19.7 | 0.43 | 256 | 241 |
| SEM | 23 | 0.1 | 0.31 | 2.2 | 0.07 | 15 | 15 |
| _P_-value | 0.72 | 0.33 | < 0.01 | 0.02 | 0.10 | 0.94 | 0.59 |

[1]Four ewes received a control well-fed and non-supplemented treatment (CTL), while five ewes received an underfed and mineral oil supplemented treatment (UFMO).

[2]Empty body: total body minus gut contents and wool. _Post-mortem_ measurements for Day+57 or estimated at Day 0 based on empty body burden measured at Day+57 plus amount excreted through faeces and wool, minus amount ingested from Days 0 to Day +57.

**Table 3. Pollutants stored in body tissues of ewes at slaughter (Day +57 of depuration period)[1].**

| | Adipose tissues | | | Liver | *Rect. abdo.* muscle[3] | Blood serum |
|---|---|---|---|---|---|---|
| | **Mesenteric** | **Perirenal** | **Pericaudal sc.[2]** | | | |
| | | | TCDD (ng) | | | |
| CTL | 14.4 | 12.2 | 1.87 | 1.63 | 0.24 | 0.04 |
| UFMO | 11.2 | 11.0 | 1.28 | 1.84 | 0.26 | 0.04 |
| SEM | 1.4 | 4.2 | 0.36 | 0.17 | 0.02 | 0.01 |
| *P*-value | 0.10 | 0.83 | 0.29 | 0.40 | 0.47 | 0.75 |
| | | | PCB 126 (ng) | | | |
| CTL | 13.9 | 12.7 | 1.74 | 4.65 | 0.23 | 0.08 |
| UFMO | 14.5 | 11.0 | 1.31 | 4.48 | 0.24 | 0.08 |
| SEM | 4.4 | 2.9 | 0.36 | 0.51 | 0.02 | 0.01 |
| *P*-value | 0.93 | 0.70 | 0.44 | 0.82 | 0.81 | 0.90 |
| | | | PCB 153 (µg) | | | |
| CTL | 14.1 | 15.5 | 1.87 | 0.93 | 0.30 | 0.04 |
| UFMO | 13.9 | 15.1 | 1.39 | 1.02 | 0.28 | 0.04 |
| SEM | 1.9 | 3.3 | 0.32 | 0.14 | 0.03 | 0.01 |
| *P*-value | 0.95 | 0.97 | 0.32 | 0.69 | 0.58 | 0.86 |

[1]Four ewes received a control well-fed and non-supplemented treatment (CTL), while five ewes received an underfed and mineral oil supplemented treatment (UFMO).

[2]Pericaudal sc.: Pericaudal subcutaneous.

[3]*Rect. abdo.*: Rectus abdominis.

and 54% of the POPs oral input during the exposure and buffering periods for TCDD, PCBs 126 and 153, respectively (Table 2). When expressed relative to the calculated empty body burdens at the beginning of the depuration period (Day 0), burdens at slaughter (Day +57)

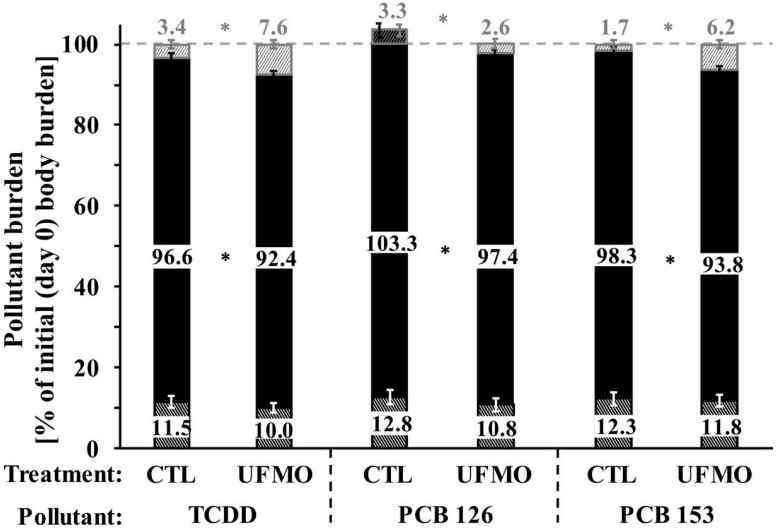

**Fig 2. Distribution and excretion balances of pollutants between adipose tissues (sum of mesenteric, perirenal and pericaudal subcutaneous, white shaded area) and empty body (■) at slaughter, and input/outputs balance during the depuration period (sum of faecal and wool outputs–feed input from Day 0 to Day +57, grey shaded area) in ewes receiving a control well-fed and non-supplemented treatment (CTL, *n* = 4) or an underfed and mineral oil supplemented treatment (UFMO, *n* = 5). Bars represent least-squares means, error bars represent SEM and * represents a difference between treatments ($P \leq 0.05$).**

decreased ($P<0.05$) only slightly from 97%, 103%, and 98% for CTL to 92%, 97%, and 94% for UFMO for TCDD, PCBs 126 and 153, respectively (Fig 2).

In absolute amounts at slaughter, only the mesenteric AT TCDD burden tended to decrease ($P = 0.10$) by 1.3-fold, whereas numerical non-significant ($P > 0.10$) 1.3- to 1.5-fold decreases were observed for pericaudal subcutaneous AT TCDD, PCBs 126 and 153 burdens due to UFMO treatment compared to CTL treatment (Table 3). However, over the depuration period, UFMO treatment greatly increased ($P \leq 0.05$) faecal TCDD, PCBs 126 and 153 excretions by 1.4- to 2.1-fold but tended to decrease ($P = 0.10$) wool PCB 153 excretion by 1.4-fold compared to CTL treatment (Table 2).

## Discussion

### Body lipid dynamics

In response to UFMO treatment, ewes mobilized their body lipid reserves, as ascertained by the decreases in BW, body condition score, and adipose cell size, which are in broad accordance with the literature [6]. The comparison of allometric coefficients revealed that lipid mobilization was more intense in perirenal and subcutaneous AT than in mesenteric AT and finally muscle, which is in agreement with data on Barbarine ewes [21].

### Fate of TCDD and PCBs in ewes

**Tissue distribution.** In the CTL ewes, POPs concentrations based on lipid weight were remarkably similar among empty body, mesenteric, perirenal and subcutaneous AT, and muscle as they did not differ by more than 1.1-, 1.2- and 1.4-fold for TCDD, PCB 126 and PCB 153, respectively. Such a pattern of tissue distribution of lipophilic POPs depending on tissue lipid contents is in accordance with the pattern previously observed for TCDD in non-lactating cows [4] and growing lambs [5] and for PCB 126 in dairy cows [24]. Conversely, blood serum TCDD and PCB 153 concentrations were 2.8- to 4.5-fold lower, and blood serum PCB 126 concentrations were 1.3- to 1.6-fold lower compared to empty body, AT and muscle concentrations. This could be explained by the fact that neutral lipids (i.e. triglycerides and cholesteryl esters), rather than total lipids are considered as the main pool of distribution for non-polar POPs [25]. Indeed, in ruminants the neutral lipids/total lipids ratio is expected to be approximately 2.5-fold lower in serum than in AT or *Rectus abdominis* muscle [26, 27]. The only exception to this pattern of distribution was the liver with 2.5- to 8-fold higher TCDD and PCB 126 concentrations in total lipids compared to empty body, AT and muscle (only 0.9- to 1.3-fold for PCB 153), while ewe liver lipids are poor in neutral lipids (< 20% of total lipids; [28]). Specific liver accumulations of similar magnitude were reported for TCDD in non-lactating cows [4] and in growing steers [3] and for TCDD and PCB 126 in growing lambs [5].

Treatment UFMO affected the distribution of POPs, with 1.2- to 1.5-fold higher concentrations in AT and muscle compared to CTL treatment. It seems that the increase in POPs concentration for a specific tissue in response to UFMO occurred to a degree that depends on the tissue lipids mass decrease over the depuration period. Indeed, in UFMO ewes, 1.2- to 1.3-fold higher POPs concentrations were observed in perirenal and subcutaneous AT than in mesenteric AT and muscle, whereas the two former tissues showed a higher allometric coefficient for lipids ($\geq 2.4$) than the two latter ($\leq 0.9$). An even 2-fold higher concentration effect was observed for PCB 153 in the pericaudal subcutaneous AT of underfed (35% of MER) ewes over a 21-day depuration period, when compared to the initial concentration observed before the underfeeding period [12]. Together, this suggests that UFMO ewes mobilized AT lipids, but did not eliminate POPs in proportional amounts. Indeed, no net changes in burdens was observed whatever the tissues in UFMO compared to CTL ewes. Conversely, in mice

undernutrition (50% of *ad libitum* intake) combined with non-absorbable lipids supplementation (10% of olestra in total diet) over a 15-day depuration period decreased by 1.8-fold the epididymal AT hexachlorobenzene burden [29]. Additionally, in pigeons complete starvation over 6 days decreased the omental AT burden of DDT by 3.9-fold but increased that of breast muscle by 1.8-fold [8]. This discrepancy between studies could be explained, at least in part, by the differential lipid metabolism adaptation in response to undernutrition between ruminants, rodents and birds. On one hand, differential intensity of lipid mobilization in response to undernutrition occurred between mice and ewes (1.4- to 1.6-fold decrease in AT lipid mass in ewes in the present study vs. 2.9-fold decrease in epididymal AT mass in mice [29]). On the other hand, lipoprotein lipase activity of oxidative muscle is known to decrease in ruminants in response to undernutrition, whereas it increases in birds [10], which could imply a resulting increase in POPs uptake of oxidative breast muscle in underfed pigeons [8], but not for the oxidative *Rectus abdominis* muscle in UFMO ewes (present study).

**Faecal and wool excretions.** Only excretions through faeces and wool were compared. Urine or breathing outputs were not explored as they were expected to contribute very little to the overall excretion of highly lipophilic [coefficient of partition between octanol and water ($K_{ow}$) higher than $10^{6.5}$] and non-volatile [coefficient of partition between octanol and air ($K_{oa}$) higher than $10^{10}$] studied compounds. In response to UFMO, faecal excretion was increased by 2.1-, 1.4- and 1.9-fold for TCDD, PCBs 126 and 153, respectively. These results obtained by pool sampling (16 individual faecal samples collected straight from the rectum every 3 to 4 days during the 8-week depuration period further pooled together per ewe) are in full agreement with those obtained from kinetic single day sampling and pooling per treatment [6]. Similarly, supplementing diets with mineral oil enhanced faecal POPs excretions by 2- to 3-fold in well-fed dairy goats contaminated by mirex [11] and in well-fed growing lambs contaminated by hexachlorobenzene [30]. Therefore, UFMO does not seem to enhance the faecal excretion of POPs more than mineral oil supplementation alone [11, 30], when indirectly compared. This result contrasts with the synergetic effect of underfeeding combined with non-absorbable lipids supplementation observed in mice [31] and growing chickens [7], a discrepancy that is discussed further in the companion paper [6].

To the best of our knowledge, the present study is the first to quantify the amounts of POPs excreted through wool. The significance of this route of elimination in the whole depuration process of ewes is slight, as wool excretion contributed less than 5% of the cumulative faecal and wool excretion of POPs, whatever the treatment. Treatment only marginally affects the wool excretion of PCB 153, which was numerically 1.4-fold lower in UFMO than in CTL ewes. This is mainly explained by the 1.7-fold decrease in wool DM (1.3-fold for lipids) growth due to UFMO. In the CTL ewes, wool had 1.2-, 2.9- and 1.8-fold higher TCDD, PCBs 126 and 153 concentrations based on lipid weight, respectively, compared to empty body, whereas in the UFMO ewes those increase were slightly lower (1.2-, 2.1- and 1.4-fold for TCDD, PCBs 126 and 153, respectively). Besides, Gill et al. [32] reported 1- to 2-fold higher PCBs concentration in wool than in subcutaneous AT of ewes reared on contaminated pasture.

**Assimilation efficiency and body burden.** An assimilation efficiency of between 48% and 54% for TCDD, and PCBs 126 and 153 was observed in the present study. This is in broad accordance with the 53% assimilation efficiency estimated for TCDD in growing steers over a 120-day contamination period [3], and is remarkably close to the 45% net absorption for TCDD measured during an input/ouptut mass balance experiment in non-lactating cows [4]. Assimilation efficiency is therefore a reliable estimate of dietary absorption rates in the case of TCDD, and PCBs 126 and 153, which is explained by the very small contribution to the overall body clearance of routes of elimination other than faecal and wool excretions (i.e. urinary and air breathing excretions, and metabolism) [33, 34].

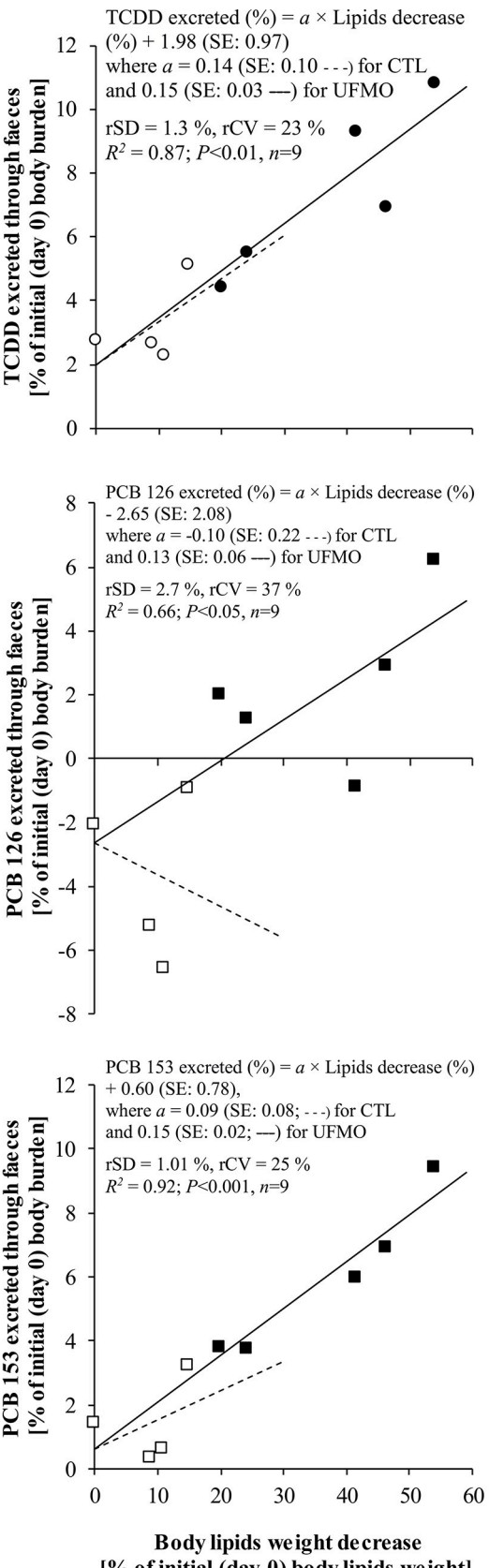

**Fig 3. Relationships between empty body lipid weight decrease and corrected amounts of pollutants excreted through faeces (faecal output–feed input) for TCDD (○, •), and PCBs 153 (□, ■) and 126 (Δ, ▲) over a 57-day**

depuration period in ewes receiving a control well-fed and non-supplemented treatments (CTL: ○, □, Δ; *n* = 4), or an underfed and mineral oil supplemented treatment (UFMO: •, ■, ▲; *n* = 5).

The cumulative faecal and wool excretions over the 57-day depuration period allowed the decrease in body burden from -3.4% and -1.7% of the initial body burden (Day 0) in CTL to -7.6% and -6.2% in UFMO ewes for TCDD and PCB 153, respectively. For PCB 126, a decrease of -2.6% was also observed in UFMO, whereas body burden increases from +3.3% in CTL ewes due to the PCB 126 background contamination of feedstuffs (especially straw and hay) offered to ewes during the depuration period. For UFMO, those decreases are far from that observed in growing chickens contaminated with PCBs, where 21 days of depuration in feed restriction (50% of *ad libitum* intake) combined with MO supplementation (10% of total diet) decreased of -67% the total PCBs body burden, compared to a control *ad libitum* and non-supplemented treatment [7]. Both the differences in POPs physico-chemical properties (lipophilicity and metabolic clearance rate), the weight of lipids excreted through faeces relative to that stored in the empty body (ratio of 0.3 in ewes vs. 2.5 in chickens) and the severity of the decrease in body lipid weight (-33% of initial lipids mass in ewes vs. -71% in chickens) due to UFMO, could explain this discrepancy between studies.

Concerning the last assumption, clear linear and positive relationships were observed in the present study between the intensity of body lipid mobilization and the rate of POPs corrected faecal excretion (POPs faecal input–POPs feed input over the depuration period, Fig 3). Considering the slope of the relationships for UFMO ewes, for a 10% decrease in initial empty body lipid weight, approximately 1.3% to 1.5% of the initial POPs body burdens were excreted through faeces over the depuration period.

## Conclusions

The results of this study showed that UFMO treatment enhanced the faecal excretion of TCDD and PCBs in non-lactating ewes by 2-fold compared to the CTL treatment. When considering faecal and wool excretion as the sole routes of elimination, a first-order kinetic of depuration applied to empty body burdens at Days 0 and +57 allows to derivate half-lives of 1 200 and 520 days for TCDD, and of 2 020 and 650 days for PCB 153, for CTL and UFMO treatments, respectively. Nonetheless, cumulative faecal excretion accounted for less than 8% of the initial body burden whatever the POPs over the 57-day depuration period, even for the UFMO ewes. In addition, AT and muscle concentrations were increased by 1.2- to 1.5-fold in UFMO ewes concomitantly with the 1.5-fold decrease in empty body lipid weight. Therefore, in case of an on-farm contamination incident involving ewes and lipophilic and poorly metabolized POPs such as TCDD and PCBs 126 and 153, a UFMO strategy may not be beneficial unless i) ewes are contaminated at a level only slightly higher than maximal regulatory limits, and ii) UFMO is completed by a refeeding period when tissue concentrations would be further reduced by the residual burden dilution in body lipids weight increase. In this instance, the reduction of the POPs body burden half-life by 2- to 3-fold could make the difference between disposal or salvage of expensive livestock. Further supporting data regarding the effect of UFMO treatment on the depuration of other POPs in different ruminant species are needed to extent broadly the results of the present study.

## Supporting information

**S1 Table. Empty body chemical composition and adipocyte volume of ewes at slaughter (end of depuration period, day +57).**
(DOCX)

**S2 Table. Weights of ewes body compartments at slaughter and of oral intakes and output compartments over the depuration period.**
(DOCX)

**S3 Table. Dioxin (TCDD) and polychlorinated biphenyls (PCBs) concentrations in body tissues of ewes at slaughter and in output compartments over the depuration period.**
(DOCX)

**S1 File. POPs analyses method.**
(DOCX)

**S2 File. Codes for statistical analyses.**
(DOCX)

## Acknowledgments

The authors thank C. Coustet, J. Mongiat and S. Collange (INRA, UE 1414) for slaughter procedures; J. Liger, J.F. Rouaud, and M. Alix (INRA, UE Porcs Rennes, Saint-Gilles, France) for empty body mincing; and P. Hartmeyer (Université de Lorraine, INRA, UR AFPA) for organ grinding, DM, and ashes determination.

## Author Contributions

**Conceptualization:** Sylvain Lerch.

**Data curation:** Sylvain Lerch, Lucille Rey-Cadilhac.

**Formal analysis:** Sylvain Lerch, Lucille Rey-Cadilhac.

**Funding acquisition:** Sylvain Lerch, Ronan Cariou, Anne Ferlay.

**Investigation:** Lucille Rey-Cadilhac, Denis Roux.

**Methodology:** Ronan Cariou, Yannick Faulconnier, Catherine Jondreville, Stefan Jurjanz.

**Project administration:** Sylvain Lerch, Lucille Rey-Cadilhac, Anne Ferlay.

**Resources:** Denis Roux, Gaud Dervilly-Pinel, Bruno Le Bizec.

**Software:** Sylvain Lerch, Lucille Rey-Cadilhac.

**Supervision:** Sylvain Lerch, Anne Ferlay.

**Validation:** Sylvain Lerch, Ronan Cariou, Anne Ferlay.

**Visualization:** Lucille Rey-Cadilhac.

**Writing – original draft:** Sylvain Lerch.

**Writing – review & editing:** Lucille Rey-Cadilhac, Ronan Cariou, Yannick Faulconnier, Catherine Jondreville, Stefan Jurjanz, Anne Ferlay.

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
