## [Decision Letter · Decision Letter 0]

9 Jan 2020

PONE-D-19-30218

Undernutrition combined with dietary mineral oil hastens depuration of stored dioxin and polychlorinated biphenyls in ewes. 2. Tissue distribution, mass balance and body burden

PLOS ONE

Dear Dr. Lerch,

Thank you for submitting your manuscript to PLOS ONE. After careful consideration, we feel that it has merit but does not fully meet PLOS ONE’s publication criteria as it currently stands. Therefore, we invite you to submit a revised version of the manuscript that addresses the points raised during the review process.

We would appreciate receiving your revised manuscript by Feb 23 2020 11:59PM. To enhance the reproducibility of your results, we recommend that if applicable you deposit your laboratory protocols in protocols.io, where a protocol can be assigned its own identifier (DOI) such that it can be cited independently in the future. For instructions see: http://journals.plos.org/plosone/s/submission-guidelines#loc-laboratory-protocols

We look forward to receiving your revised manuscript.

Kind regards,

Juan J Loor

Academic Editor

PLOS ONE

Journal Requirements:

Reviewers' comments:

Reviewer's Responses to Questions

**Comments to the Author**

1. Is the manuscript technically sound, and do the data support the conclusions?

Reviewer #1: Yes

2. Has the statistical analysis been performed appropriately and rigorously? 

Reviewer #1: Yes

3. Have the authors made all data underlying the findings in their manuscript fully available?

Reviewer #1: Yes

4. Is the manuscript presented in an intelligible fashion and written in standard English?

Reviewer #1: No

5. Review Comments to the Author

Reviewer #1: In the submitted manuscript the authors investigated the efficacy of a depuration protocol, including undernutrition and mineral oil administration, in ewes dietary exposed to TCDD, PCB126 and PCB153. Chemical analyses were performed in different body tissues, as well as in feces and wool.

The study is well designed and performed with appropriate techniques. However, there are some minor concerns that should be considered before publishing:

1. Why did the authors select the investigated congeners? An explanation for their choice, which includes 2 DL-compounds and 1 NDL, would be noteworthy.

2. The paper is quite well written; however, to be suitable for publication, the manuscript should undergo English editing by a native English speaker.

3. Lines 221-224: please, rephrase since the sentence is too long and hard to read.

4. Table 1: please check the values of the “changes during depuration period”, since in some cases they are not correct.

5. Lines 284-288: the higher accumulation of DL-compounds (and not NDL PCBs) in sheep liver compared to other species, including other ruminants, is well known. However, the reason for such phenomenon is likely not related to higher hepatic CYP1A levels, since in ovine they are lower than in bovine. In my opinion the authors should revise this part of the discussion, providing other references.

6. Lines 290-291: In my opinion the sentence is not clear. Please, rephrase.

7. Lines 388-395: the authors drew a general conclusion that refers broadly to all POPs and all livestock species. Although their data are interesting and robust, in my opinion they should be cautious in extending their results to other species and POPs, due to the well-known kinetics differences. Accordingly, in the abstract (lines 29 and 43), “lipophilic food contaminants” are too generic.

6. PLOS authors have the option to publish the peer review history of their article (what does this mean?). If published, this will include your full peer review and any attached files.

Reviewer #1: No

---

## [Author Response · Author response to Decision Letter 0]

14 Feb 2020

Subject: Manuscript PONE-D-19-30218: “Undernutrition combined with dietary mineral oil hastens depuration of stored dioxin and polychlorinated biphenyls in ewes. 2. Tissue distribution, mass balance and body burden“

Dear Editor,

We appreciate all of the constructive comments and thoughtful suggestions provided by the referee. All suggestions arising from the peer-review have been carefully considered during revision of the original manuscript. In all cases, the suggested changes have been carefully considered and implemented in full.

The associate changes are highlighted in red thorough the manuscript labeled 'Revised Manuscript with Track Changes’. A response to the comment from the referee is outlined in the accompanying response indicated by sentences starting with AU:. The line number reported in the response are the one of the 'Revised Manuscript with Track Changes’.

We hope that this revision would allow the manuscript to be considered acceptable for publication.

With our best regards,

S. Lerch,

 

Reviewers Comments to the Author

Reviewer #1: In the submitted manuscript the authors investigated the efficacy of a depuration protocol, including undernutrition and mineral oil administration, in ewes dietary exposed to TCDD, PCB126 and PCB153. Chemical analyses were performed in different body tissues, as well as in feces and wool.

The study is well designed and performed with appropriate techniques. However, there are some minor concerns that should be considered before publishing:

1. Why did the authors select the investigated congeners? An explanation for their choice, which includes 2 DL-compounds and 1 NDL, would be noteworthy.

AU: We agree that no justification about the choices of these three specific molecules among the dioxins and PCBs families was included in the introduction section of the initial submission. A sentence was added at the end of the introduction section in order to justify this choice. See lines lines 62-65.

2. The paper is quite well written; however, to be suitable for publication, the manuscript should undergo English editing by a native English speaker.

AU: The R1 version of the manuscript was subjected to an additional English editing step provide by the Scribendi services (https://www.scribendi.com/). See the concomitant changes along the main text highlighted in green.

3. Lines 221-224: please, rephrase since the sentence is too long and hard to read.

AU: The sentence was split in two and rewrite accordingly. See lines 234-238.

4. Table 1: please check the values of the “changes during depuration period”, since in some cases they are not correct.

AU: Thank you for notice this. The lsmeans and P value of the model for empty body lipids mass at day +57 were not correct initially. Table 1 was corrected accordingly. Moreover, in the case of pericaudal adipose cell diameter data for the CTL treatment: difference between lsmeans reported for day 0 and 57 is of -3 µm, but a value of -4 µm is reported for the delta model. This is simply due to the rounding of the data (exact results are -3.75 µm). See the concomitant changes throughout the Table 1.

5. Lines 284-288: the higher accumulation of DL-compounds (and not NDL PCBs) in sheep liver compared to other species, including other ruminants, is well known. However, the reason for such phenomenon is likely not related to higher hepatic CYP1A levels, since in ovine they are lower than in bovine. In my opinion the authors should revise this part of the discussion, providing other references.

AU: We agree and further remove this sentence. Now, we only state the consistency of the results of our study when compared to previous one reporting dioxins and PCBs liver concentrations in bovine and ovine. We did not provide any more putative mechanistic explanations for liver sequestration, as the results of our study did not allow us to support any specific hypotheses. See lines 299-302.

6. Lines 290-291: In my opinion the sentence is not clear. Please, rephrase.

AU: The sentence has been rewritten accordingly the comment. See lines 304-306.

7. Lines 388-395: the authors drew a general conclusion that refers broadly to all POPs and all livestock species. Although their data are interesting and robust, in my opinion they should be cautious in extending their results to other species and POPs, due to the well-known kinetics differences. Accordingly, in the abstract (lines 29 and 43), “lipophilic food contaminants” are too generic.

AU: We agree and further adjusted the concluding statements. See lines 405-415. Moreover, terminology was reviewed throughout the text, retaining only “persistent organic pollutants” and no more “lipophilic contaminants”. See concomitant changes thorough the abstract and the text.

---

## [Decision Letter · Decision Letter 1]

5 Mar 2020

Undernutrition combined with dietary mineral oil hastens depuration of stored dioxin and polychlorinated biphenyls in ewes. 2. Tissue distribution, mass balance and body burden

PONE-D-19-30218R1

Dear Dr. Lerch,

We are pleased to inform you that your manuscript has been judged scientifically suitable for publication and will be formally accepted for publication once it complies with all outstanding technical requirements.

With kind regards,

Juan J Loor

Academic Editor

PLOS ONE

Additional Editor Comments (optional):

Reviewers' comments:

Reviewer's Responses to Questions

**Comments to the Author**

1. If the authors have adequately addressed your comments raised in a previous round of review and you feel that this manuscript is now acceptable for publication, you may indicate that here to bypass the “Comments to the Author” section, enter your conflict of interest statement in the “Confidential to Editor” section, and submit your "Accept" recommendation.

Reviewer #1: All comments have been addressed

2. Is the manuscript technically sound, and do the data support the conclusions?

Reviewer #1: Yes

3. Has the statistical analysis been performed appropriately and rigorously? 

Reviewer #1: Yes

4. Have the authors made all data underlying the findings in their manuscript fully available?

Reviewer #1: Yes

5. Is the manuscript presented in an intelligible fashion and written in standard English?

Reviewer #1: Yes

6. Review Comments to the Author

Reviewer #1: (No Response)

7. PLOS authors have the option to publish the peer review history of their article (what does this mean?). If published, this will include your full peer review and any attached files.

Reviewer #1: No

---

## [Editor Report · Acceptance letter]

16 Mar 2020

PONE-D-19-30218R1 

Undernutrition combined with dietary mineral oil hastens depuration of stored dioxin and polychlorinated biphenyls in ewes. 2. Tissue distribution, mass balance and body burden 

Dear Dr. Lerch:

I am pleased to inform you that your manuscript has been deemed suitable for publication in PLOS ONE. Congratulations! Your manuscript is now with our production department. 

With kind regards,

on behalf of

Dr. Juan J Loor 

Academic Editor

PLOS ONE